

# Multiyear statistics of columnar ice production in stratiform clouds over Hyytiälä, Finland

Haoran Li[1], Ottmar Möhler[2], Tuukka Petäjä[1], and Dmitri Moisseev[1,3]

[1]Institute for Atmospheric and Earth System Research / Physics, Faculty of Science, University of Helsinki, Finland
[2]Institute of Meteorology and Climate Research, Karlsruhe Institute of Technology, Karlsruhe, Germany
[3]Finnish Meteorological Institute, Helsinki, Finland

**Correspondence:** Haoran Li (haoran.li@helsinki.fi)

**Abstract.** Formation of ice particles in clouds at the temperatures of -10 °C or warmer was documented by using ground-based remote sensing observations. At these temperatures, the number concentration of ice nucleating particles (INP) is not only expected to be small, but also this number is highly uncertain. In addition, there are a number of studies reporting that the observed number concentration of ice particles exceeds expected INP concentrations, indicating that other ice generation mechanisms, such as secondary ice production (SIP), may play an important role in such clouds. To identify the formation of ice crystals and report conditions in which they are generated, W-band cloud radar Doppler spectra observations collected at the Hyytiälä station for more than two years were used. Given that at these temperatures ice crystals grow mainly as columns, which have distinct linear depolarization ratio (LDR) values, spectral LDR was utilized to identify newly formed ice particles.

Our results indicate that that the columnar ice production took place in 5 to 13 % of clouds, where cloud top temperatures were -12 C or higher. For colder clouds, this percentage can be as high as 33 %. 40 ∼ 50 % of columnar-ice-producing events last less than 1 hour, while 5 ∼ 15 % can persist for more than 6 hours. By comparing clouds where columnar crystals were produced with the ones where these crystals were absent, we found that the columnar-ice-producing clouds tend to have larger values of liquid water path and precipitation intensity. The columnar-ice-producing clouds were subdivided into subcategories, using the temperature difference, $\Delta T$, between the altitudes where columns are first detected and the cloud top altitude. The cases where $\Delta T$ is less than 2 °C are typically single-layer shallow clouds where needles are produced at the cloud top. In multilayered clouds, where $\Delta T > 2$ °C, columns are produced in a layer that is seeded by ice particles falling from above. This classification allows to study potential impacts of various SIP mechanisms, such as Hallet-Mossop process or freezing breakup, on columnar ice production. To answer the question whether the observed ice particles are generated by SIP in the observed single-layer shallow clouds, ice particle number concentrations were retrieved and compared to several INP parameterizations. It was found that the ice number concentrations tend to be 1 ∼ 3 orders of magnitude higher than the expected INP concentrations.

## 1 Introduction

Ice production in mixed-phase clouds is critical for their radiative (Sun and Shine, 1994) and microphysical (Korolev et al., 2017) properties. At temperatures warmer than -38 °C, ice crystals form on ice nucleating particles (INPs). In-situ measure-





ments have revealed that the number concentration of available INPs steeply decreases with the increase of ambient temperature. This dependence is more or less universal and seems to depend only marginally on other factors such as the geographic location, airmass types and aerosol compositions (DeMott et al., 2010; Wilson et al., 2015; DeMott et al., 2016; Petters and Wright, 2015). Above -10 °C, the typical concentrations of INPs are below $10^{-1}$ per Liter ($L^{-1}$) and can be as low as $10^{-6}$ $L^{-1}$

(Petters and Wright, 2015; Kanji et al., 2017). A number of studies, however, have reported that the ice number concentration in clouds with the top temperature warmer than -10 °C can exceed the expected concentration of INPs by several orders of magnitude (e.g., Mossop, 1985; Hobbs and Rangno, 1985; Rangno and Hobbs, 2001). This discrepancy implies that numerical weather prediction models that rely solely on INP parameterizations cannot realistically represent ice number concentrations in moderately to slightly supercooled clouds. As a result, the inappropriate parameterization of ice production may lead to biased

estimates of surface shortwave radiation budget (Young et al., 2019), among other things.

Several mechanisms have been proposed to explain this discrepancy, such as the enhanced contact nucleation driven by the thermophoretic force during the evaporation of liquid drops (Beard, 1992; Hobbs and Rangno, 1985), pre-activated INPs from evaporated ice particles nucleated above (Roberts and Hallett, 1968; Fridlind et al., 2007) or secondary ice production (SIP) mechanisms (see recent reviews by Field et al., 2017; Korolev and Leisner, 2020). The SIP has been studied by a number of

laboratory experiments since 1940s (e.g., Findeisen and Findeisen, 1943; Dye and Hobbs, 1966; Wildeman et al., 2017). Hallett and Mossop (1974) have found that numerous ice splinters can be generated when supercooled liquid drops with the diameter larger than $\sim$ 24 $\mu m$ are collected by large ice particles within the temperature range of -8 °C $\sim$ -3 °C. This is referred as to Hallett–Mossop (H-M) process, the most studied and the only implemented secondary ice production (SIP) mechanism in numerical cloud, weather, and climate models (Morrison et al., 2005). The enhanced ice number concentration can also be

caused by the fragmentation of large supercooled liquid droplets (e.g., Evans and Hutchinson, 1963; Scott and Hobbs, 1977; Wildeman et al., 2017). It has been found that the secondary ice production efficiency is positively correlated with the size of liquid droplets (Lauber et al., 2018) and is enhanced in moist environment (Keinert et al., 2020). At temperatures higher than -3 °C the fragmentation of drizzle is still active as shown by field observations (Lauber et al., 2021). In addition, studies using optical sensors mounted on aircrafts have reported the high concentration of ice columns within the temperature range of -10

$\sim$ -3 °C (e.g., Koenig, 1963; Hobbs and Rangno, 1990; Rangno and Hobbs, 2001). Based on aircraft measurements from two field campaigns, Korolev et al. (2020) have concluded that the secondary ice process is highly associated with the presence of liquid droplets and aged rimed ice in turbulent regions. Yang et al. (2020) have found that the ice concentration in tropical maritime stratiform clouds characterized by the top temperature above -8 °C is in the order of $10^{-1}$ $\sim$ $10^{1}$ $L^{-1}$, which cannot be fully explained by primary ice nucleation, H-M process or droplet freezing. However, despite the advantage of offering a

direct way of interpreting ice microphysics, aircraft observations are only available from a few measurement campaigns, and do not seem sufficient for a long-term assessment.

The polarimetric variables, such as differential reflectivity ($Z_{dr}$), specific differential phase measurements ($K_{dp}$) and linear depolarization ratio (LDR), observed by dual-polarization radars are sensitive to the shape of hydrometeors (Bringi and Chandrasekar, 2001) and allow the analysis of ice particles with specific habits (e.g., Matrosov et al., 2001; Hogan et al., 2002;

Tyynelä and Chandrasekar, 2014; Li et al., 2018). At temperatures of -10 $\sim$ -2 °C, the depositional growth of an ice crystal



is stronger at the basal faces than at the prism faces. Hence, the formation of columnar ice is preferred (Lamb and Verlinde, 2011). This distinct habit can produce high $Z_{dr}$ and $K_{dp}$ as observed by dual-polarization weather radars (Hogan et al., 2002; Giangrande et al., 2016; Sinclair et al., 2016). For vertically-pointing Ka- and W-band radars, ice columns usually produce LDR values as high as -15 dB, which is distinctively higher than that of most other ice particle types (Aydin and Walsh, 1999; Tyynelä et al., 2011). Oue et al. (2015); Li and Moisseev (2020); Luke et al. (2021) have shown that this strong LDR signal at the slow falling part of radar Doppler spectrum can be used to identify columnar ice crystals. In this study, this method is applied to long-term radar Doppler spectra observations for characterizing the production of columnar ice particles in stratiform clouds. Similar to (Luke et al., 2021) we show that this phenomenon is not uncommon. By comparing radar-based retrievals of ice number concentrations to INP parameterizations, one of which was derived from observations collected at our measurement site (Schneider et al., 2020), we show that the ice number concentrations tend to be $1 \sim 3$ orders of magnitude higher than the expected INP concentration. This also supports the conclusions reached by Luke et al. (2021).

The paper is organized as follows. Section 2 introduces the data used in this study. The method for identifying columnar ice particles from radar Doppler spectra is illustrated in Sect. 3. Statistical results are presented in Sect. 4. Section 5 compares the concentrations of columnar ice particles and INPs in single-layer shallow clouds. Conclusions are given in Sect. 6.

## 2 Data

### 2.1 Cloud radar observations

The measurements used in this study were collected at Station for Measuring Ecosystem - Atmosphere Relations II (Hari and Kulmala, 2005; Petäjä et al., 2016) located in Hyytiälä, Southern Finland (61.845°N, 24.287°E, 150 m above mean sea level, amsl.). Since November 2017, a 94-GHz dual-polarization frequency-modulated continuous-wave Doppler cloud radar (Küchler et al., 2017) (HYytiälä Doppler RAdar, HYDRA-W) has been operating at the station. The radar is pointing vertically and measures radar signal spectral moments, linear depolarization ratio (LDR) and dual-polarization Doppler spectra, see (Li and Moisseev, 2020) for the example of the data. The radar operates using three chirps that define range resolution, Doppler unambiguous velocity and spectral resolution at three range intervals. Between 102 m and 996 m, the range resolution is 25.5 m, Doppler unambiguous velocity is 10.24 $m\ s^{-1}$ and the Doppler spectral resolution is 0.02 $m\ s^{-1}$. Between 996 and 3577 m, these values are 25.5 m, 5.12 $m\ s^{-1}$ and 0.02 $m\ s^{-1}$, respectively. For ranges above 3577 m, they are 34 m, 5.12 $m\ s^{-1}$ and 0.02 $m\ s^{-1}$, respectively. In this study, HYDRA-W observations recorded between February 2018 and April 2020 were utilized (Moisseev, 2020).

To remove noise from Doppler spectra observations, the spectral lines with the signal-to-noise ratio less than 5 dB were filtered out. Since both co-polar and cross-polar observations were used, i.e. to compute spectral LDR, this filtering could result in complete removal of the cross-polar signal, the power of which is typically $15 \sim 30$ dB lower than that of the co-polar signal (Bringi and Chandrasekar, 2001; Moisseev et al., 2002). In such cases, no LDR values were computed.

The antenna diameter of HYDRA-W is 0.5 m which translates to the Fraunhofer far-field distance of 157 m (Sekelsky, 2002; Falconi et al., 2018) and antenna beam width of 0.56 °. Therefore, the lowest radar range bin, which is not affected by



the near-field effect is 179 m (the fourth range bin). Data recorded at distance in the radar far-field were used in this study, therefore limiting the lowest altitude to 179 m where observations were taken.

In addition to the active remote sensing system, HYDRA-W is capable of estimating the liquid water path (LWP) by using the 89 GHz passive microwave channel observations. The brightness temperature at this band is regularly calibrated using liquid nitrogen. The site specific relation between the measured 89 GHz brightness temperature and LWP was derived by the radar manufacturer using radio-sounding and reanalysis data.

## 2.2   Model temperature and humidity profiles

To obtain information on atmospheric state during the cloud observations, the output from Icosahedral Nonhydrostatic (ICON) model (Zängl et al., 2015) was used. The ICON model output is provided over all Aerosol, Clouds and Trace Gases Research Infrastructure (ACTRIS) cloud profiling stations (available at http://devcloudnet.fmi.fi/). The data has hourly temporal resolution, and the height resolution decreases with the increase of altitude, for example the height resolution is 0.16 km at the altitude of 1 km. Its atmospheric products, such as temperature, relative humidity (RH) and pressure, over Hyytiälä were interpolated into the temporal and spatial resolutions of HYDRA-W.

## 3   Methods

The mean Doppler velocity (MDV) of hydrometeors observed by a vertically pointing radar is the combination of particle terminal fall velocities and the vertical component of air motion. While Doppler velocity alone could be used to identify certain types of particles (Mosimann, 1995; Kneifel and Moisseev, 2020), there are associated limitations. These limitations include, uncertainties in hydrometeor classification due to similarities of terminal fall velocities of different particles (Locatelli and Hobbs, 1974; Barthazy and Schefold, 2006; Li et al., 2020), presence of a mixture of ice particle populations within the radar volume (Zawadzki et al., 2001; Kalesse and Luke, 2019; Li and Moisseev, 2020) and impact of air motion on the observed MDV (Protat and Williams, 2011). By using radar Doppler spectra, instead of MDV, contributions from different particle populations can be separated (e.g., Zawadzki et al., 2001; Kalesse and Luke, 2019; Radenz et al., 2019; Li and Moisseev, 2020; Luke et al., 2021). In radar Doppler spectral power observations, the presence of multiple populations of hydrometeors, such as the co-existence of supercooled liquid water and ice (e.g., Zawadzki et al., 2001; Shupe et al., 2004; Luke et al., 2010; Kalesse et al., 2016; Li and Moisseev, 2019), a mixture of different ice types (e.g., Zawadzki et al., 2001; Kalesse and Luke, 2019; Radenz et al., 2019; Li and Moisseev, 2020), could manifest as multiple spectral peaks. Even in such cases, however, classification of these particles can be ambiguous. To further improve the hydrometeor identification, dual-polarization radar observations can be used. For slant measurements the spectral differential reflectivity has been found to be useful (Spek et al., 2008). Because hydrometeors typically do not have a preferred azimuth orientation, the differential reflectivity is not very useful for the classification purposes at vertical incidence. Nonetheless, LDR can be used to identify prolate particles, such as ice columns (Oue et al., 2015). At the elevation angle of 90°, columnar ice particles can produce LDR signals as high as -16 ∼ -13 dB (Oue et al., 2015), while other ice particles may produce values smaller than -20 dB (e.g., Tyynelä et al., 2011).





Furthermore, given the relatively small fall velocities of newly produced columns, in regions where they coexist with other ice particles they usually populate the slow falling part of the Doppler spectrum. Therefore, the slow falling part characterized by high spectral LDR ($\sim$ -15 dB) in the Doppler spectrum indicates presence of columnar ice particles (Oue et al., 2015).

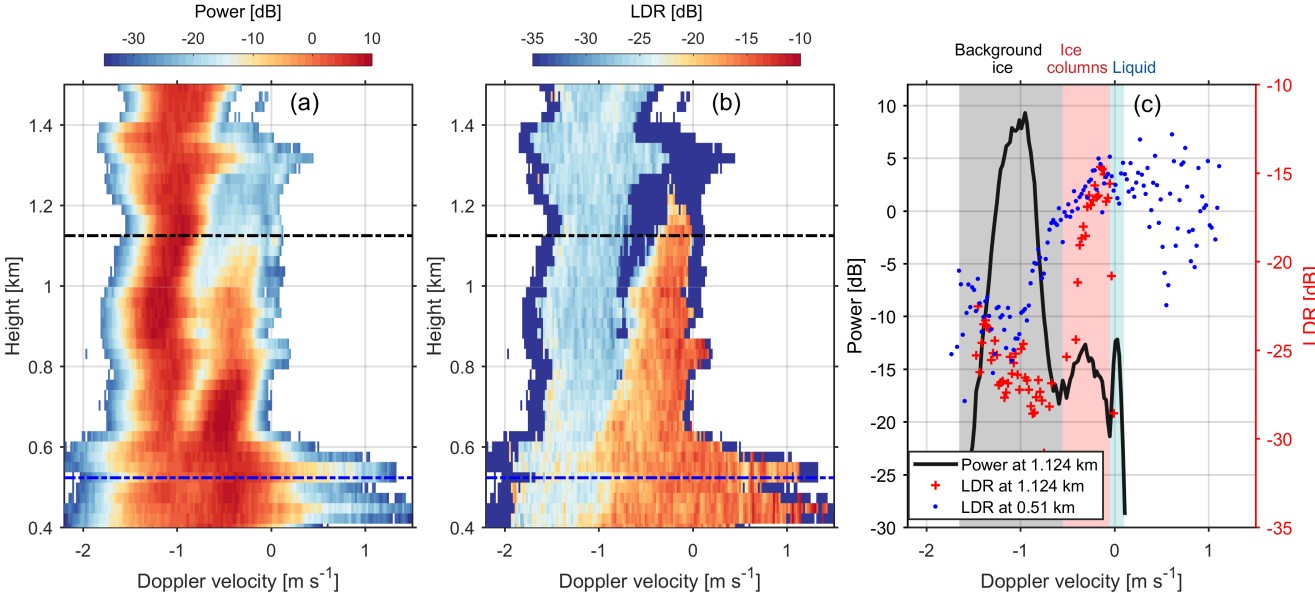

**Figure 1.** HYDRA-W Doppler spectral (a) power and (b) LDR at 2nd February 2018 07:56:38 UTC. (c) spectral power (black line) and LDR (red crosses) at 1.124 km as marked by the black dot dashed lines in (a) and (b), and LDR (blue dots) at 0.51 km as marked by the blue dot dashed lines in (a) and (b). The gray, red and blue shading areas in (c) indicate background ice falling from above, newly-formed ice columns and supercooled liquid, respectively, at 1.124 km. Negative velocity indicates downward motion downwards.

Figure 1 shows an example of observed spectral power and LDR. Two distinct populations of ice particles can be clearly identified from Fig. 1 (a), and the slower falling one corresponds to ice columns as indicated by the high spectral LDR (Fig. 1b). The observed spectral power and LDR at 1.124 km (black dot dashed lines) are shown in Fig. 1 (c). Three distinct peaks can be identified from the spectral power, and the slower falling ice columns are well characterized by the spectral LDR exceeding -18 dB. In contrast, the spectral LDR of faster falling ice is around -25 dB which mainly depends on the cross-coupling between the polarization channels (Moisseev et al., 2002) and can be much higher than the LDR signal of larger aggregates (Tyynelä et al., 2011). Interestingly, supercooled liquid water seems also present, as indicated by the well defined spectral peak at around $0\ m\ s^{-1}$ (Zawadzki et al., 2001; Shupe et al., 2004; Luke et al., 2010; Kalesse et al., 2016; Li and Moisseev, 2019). It appears that this liquid layer extends from $\sim$ 0.9 km to $\sim$ 1.3 km (Fig. 1a). The potential mechanisms of producing these ice columns will be discussed in more detailed in following sections.

Given the spectral characteristics of ice columns as discussed above, the following criteria were set to identify ice columns in clouds:





- Within the slowest 1 $m\ s^{-1}$ of the Doppler spectrum at least 3 spectral bins exceed the LDR of -18 dB.

- The temperature of the radar range bin is between -10 °C and 0 °C.

The observed radar Doppler spectrum is not only dependent on the scattering properties of hydrometeors in the radar volume, but also is affected by the turbulent broadening (Kollias et al., 2011; Tridon and Battaglia, 2015). For example, the air at around
0.51 km seems rather turbulent as indicated by the spectral power (blue dashed line in Fig. 1a). However, it appears that this issue does not significantly affect the results of columnar ice detection. The noisy spectral LDR values (blue dots in Fig. 1c) between 0.3 $m\ s^{-1} \sim 1\ m\ s^{-1}$ are attributed to the low signal-to-noise ratio. Such weak impact on spectral LDR due to turbulence may be explained by the distinctively high LDR values of ice columns, which contrast to much weaker LDR signals of ice aggregates (Tyynelä et al., 2011).

**4   Results**

By utilizing HYDRA-W Doppler spectral observations recorded between February 2018 and April 2020, statistics of environmental conditions associated with columnar ice production were derived. All detected cloud cases within the temperature range of -10 $\sim$ 0 °C were analyzed. From the selected events the cases where significant inversion was detected, which could cause melting (e.g., Kumjian et al., 2020), were excluded. Given the data selection criteria, no rainfall or summer cloud cases were
analyzed. This was done to avoid potential problems due to radar signal attenuation in rain and melting layer (Li and Moisseev, 2019). In total, 175 days of observations satisfying the data selection criteria were identified and analyzed.

**4.1   Temperature and RH conditions in columnar-ice-producing regions**

Formation and growth of ice particles requires favorable environmental conditions. These conditions were assessed by using ICON forecasts, which supplemented the radar observations. Here, we define $H_{\text{column\_top}}$ as the height where ice columns are
first detected, and $T_{\text{column\_top}}$ the temperature at this height. As shown in Fig. 2 (a), $T_{\text{column\_top}}$ values are mostly between -8 °C and -3 °C with the highest frequency at around -5 °C and a median value of -4.7 °C. There values are within the growth region of ice columns (Lamb and Verlinde, 2011). Such temperature distribution also bears a good resemblance to the results obtained from the early rime-splintering laboratory experiment (Hallett and Mossop, 1974) and recent statistical study (Luke et al., 2021). The statistics of humidity relative to ice (RH$_{\text{ice}}$) and water (RH$_{\text{liquid}}$) at $H_{\text{column\_top}}$ are shown in Fig. 2 (b). The
median values of RH$_{\text{ice}}$ and RH$_{\text{liquid}}$ are 102.6 % and 98.3 %, respectively, indicating the supply of water vapor is sufficient for growth of ice particles. This finding is, however, not surprising, since the method detects ice columns and they are growing in this temperature regime. So we can treat the presented statistics in Fig. 2 as a "sanity" check for our method.

It should be noted that although ice columns can be detected by our method, $H_{\text{column\_top}}$ may be lower than the height where ice crystals are generated. There are two potential reasons for this. Firstly, the newly formed ice particles may be spherical
(Korolev et al., 2020), and in this case they will have LDR values which are much smaller than our detection threshold. Secondly, at early stages of growth the radar signal of ice crystals is rather weak and does not allow accurate detection and



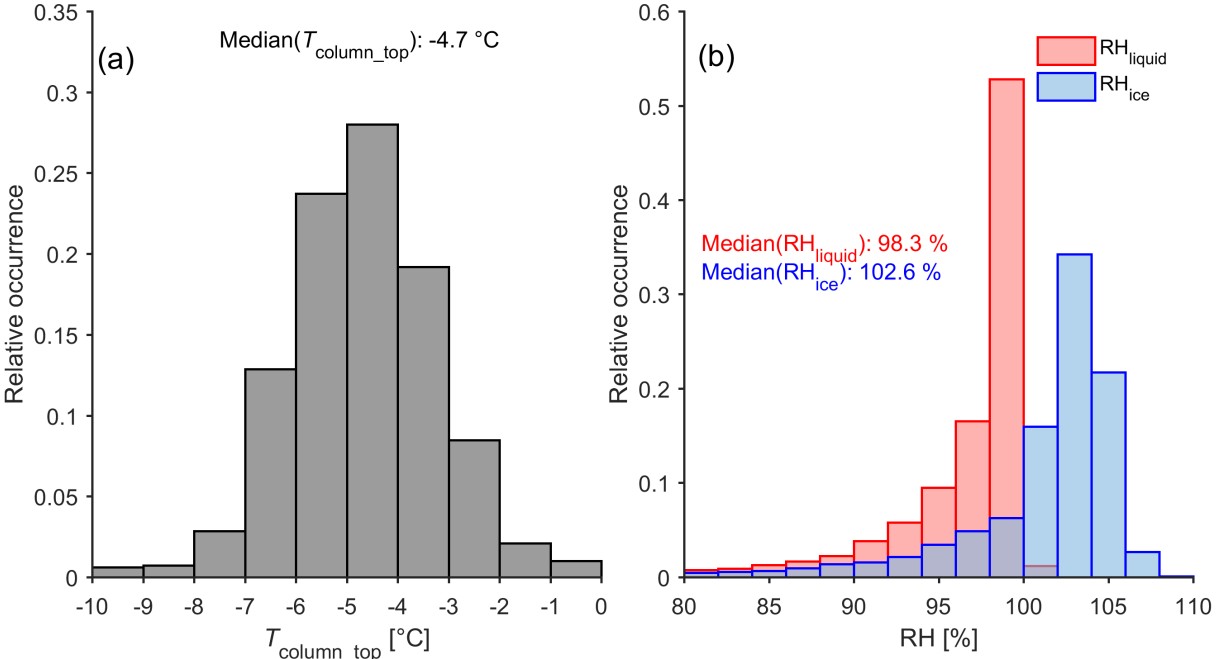

**Figure 2.** Statistics of (a) temperature and (b) relative humidity at $H_{\text{column\_top}}$ for all identified columnar-ice-producing cases.

identification of columns (Luke et al., 2021). However, the altitude difference between $H_{\text{column\_top}}$ and the actual height where columns are generated is expected to be small and not significantly affect our results.

## 4.2 Properties of columnar-ice-producing clouds

There are a number of questions that are associated with formation of ice crystals in clouds at these relatively warm temper-
5 atures. Above -10 °C, the number concentration of INPs is expected to be small and rather uncertain (DeMott et al., 2010; Kanji et al., 2017). Therefore, it is important to know how often and in which conditions these ice crystals form. Because ice formation could be facilitated by ice particles falling from upper cloud layers, i.e. by SIP, the location where ice columns are forming, with respect to the cloud top, should be identified. Finally, the importance of the columnar ice production on surface precipitation should also be assessed.

To identify such conditions cloud top temperature (CTT), defined as the temperature of the highest detected radar return for a given measurement time, is used. Because there are cases where several cloud layers are observed and there are gaps between these layers, typically the top of the lowest one is used. However, particles forming in the upper clouds, while not detected by the radar, may seed lower cloud layers, and therefore modify their properties (Vassel et al., 2019). To limit the impact of such conditions on our analysis, following (Seifert et al., 2009) we have used radar echo gap of 2 km as the threshold which defines
if the layers are connected. Recently, Proske et al. (2020) have suggested that the threshold of 2 km may overestimate the cases

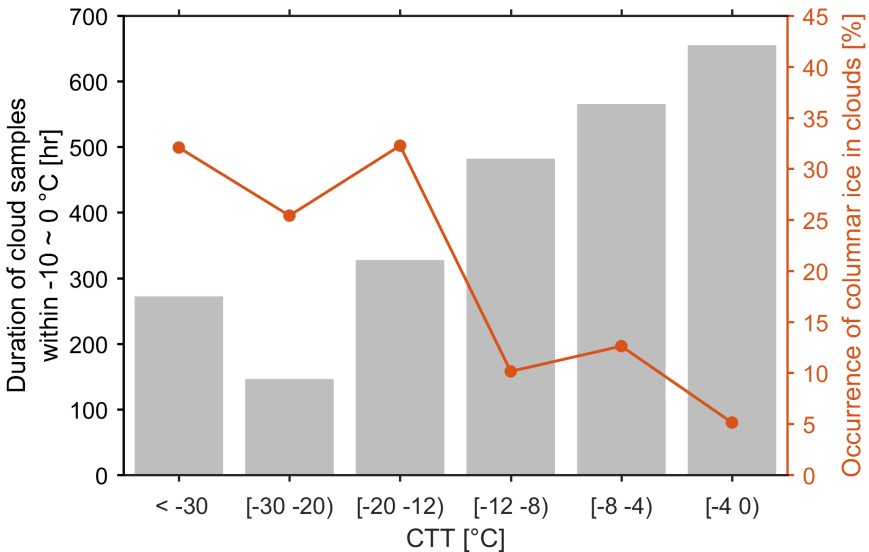

**Figure 3.** Duration of cloud observations (bars) and occurrence of columnar-ice-producing clouds (red dotted curve) over Hyytiälä as a function of CTT. The results were calculated based on the data collected from February 2018 to April 2020.

of cloud seeding. For this reason, we have also tried the threshold of 0.5 km to determine the cloud top, but we did not see significant changes in the results.

The statistics of the recorded cloud top temperatures are presented in Fig. 3. The figure (left) shows duration of detected cloud samples within the temperature of -10 ~ 0 °C for a given CTT range as recorded during the observation period. Because of the focus on cold cloud cases, where the temperature in an atmospheric column does not exceed 0 °C, the observed cloud cases were recorded between October and April. The observations show that low-level clouds, i.e. clouds with warmer CTT, are relatively more frequent. This resembles the cloud occurrence statistics of (e.g., Hagihara et al., 2010; Shupe et al., 2011). It appears deeper clouds, i.e. where CTT is below -12°C, are more conducive to columnar ice production. In these cases the frequency of columnar ice occurrence is about 25 ~ 33 %. For warmer clouds the frequency is lower and is around 5 ~ 13 %. The average occurrence is 15 %. Interestingly, our results are comparable with a recent study by Luke et al. (2021) who have found that the occurrence of columnar ice over an Arctic site is between 10 % and 25 % depending on the temperature.

As shown in Fig. 2b, the majority of columnar ice production cases took in areas of high supersaturation, which is potentially favorable for liquid water droplet formation or existence. Although direct observations of liquid were not available, the measurements collected by the 89 GHz passive channel in HYDRA-W allows estimation of LWP. The LWP values for the cloud cases are shown in Fig. 4. The observations show a significant amount of supercooled liquid water present in the atmospheric column. The cloud cases where ice columns were detected tend to have larger LWP values, especially where CTT values were smaller than -12°C. This potentially indicates that supercooled liquid droplets may play important roles in formation of ice columns. Given the mixed-phase cloud conditions, the observed columns are most probably ice needles.

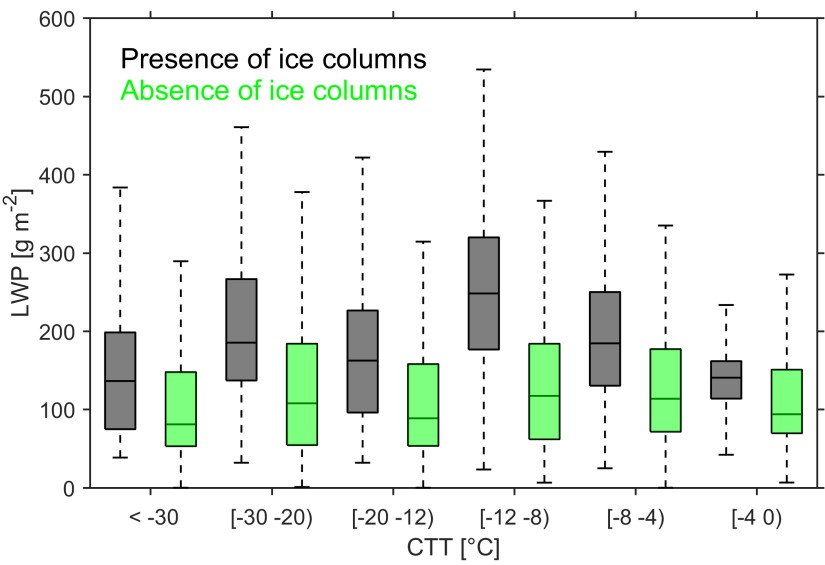

**Figure 4.** Comparison of LWP for clouds with and without columnar ice production. A cloud sample was identified if the cloud base was within the temperature range of -10 ∼ 0 °C. The boxplots represent the median (horizontal strip) and 5-95 % quantile range (whisker) of the distribution.

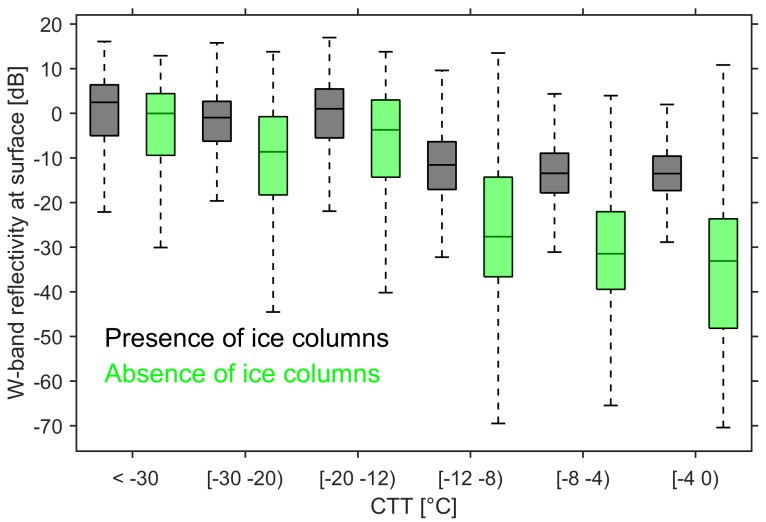

**Figure 5.** The same as Fig. 4 but for W-band radar reflectivity at the fourth range bin (179 m above the surface).

Formation of ice crystals is an efficient precipitation process (Lamb and Verlinde, 2011). To evaluate the impact of columnar ice production on precipitation, the radar equivalent reflectivity factor is used as the proxy for the precipitation intensity. For





clouds where the radar echo extends to the ground, the reflectivity values at the fourth radar gate, 179 m above ground level (agl), were used. As shown in Fig. 5 the reflectivity increases with decreasing CTT. This is due to the link between the cloud depth and precipitation intensity. The columnar ice production tends to increase the precipitation intensity. This effect is more pronounced for warmer clouds, where CTT is -12 °C or warmer. In warmer clouds the precipitation intensity can be enhanced

by as much as 10 fold. The factor of 10 increase in precipitation rate appears from the 10 dB increase in reflectivity (Falconi et al., 2018). As will be discussed in the next section, the warmer clouds tend to be single-layer clouds, where the crystal formation is more directly linked to precipitation formation. The colder clouds are prone to be consisting of the multiple cloud layers where precipitation processes are more complex.

## 4.3   Columnar ice production in single-layer and multilayered clouds

For all detected columnar-ice-producing cases, the distribution of CTT was analysed. As shown in Fig. 6 (a), ice columns can form in clouds with a wide range of CTT. The majority of cases fall in the CTT range of $-20 \sim 0°C$ with two peaks at around -15 °C and -5 °C, respectively. The peak at around -15 °C agrees rather well with the high occurrence of ice columns in clouds with the CTT of $-20 \sim -12°C$ (Fig. 3).

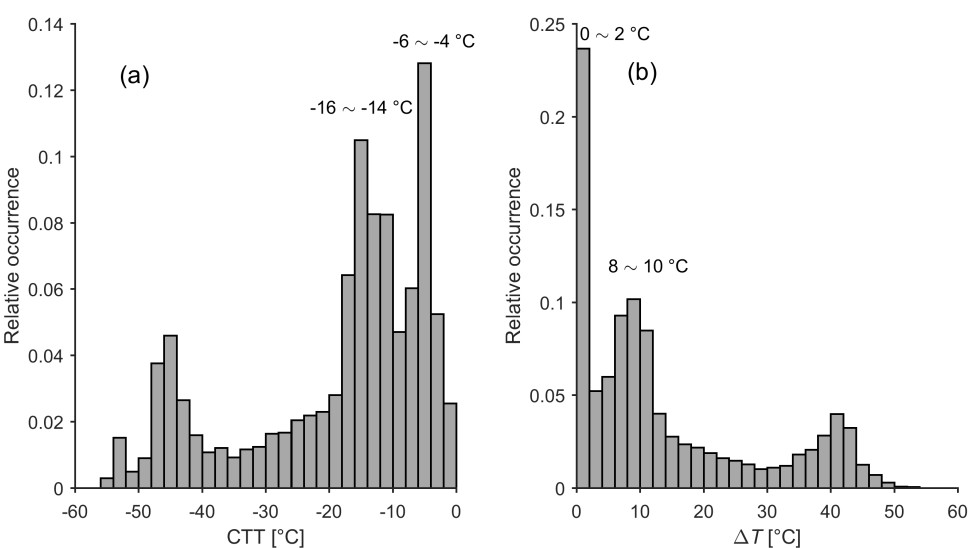

**Figure 6.** Relative occurrence of (a) CTT and (b) $\Delta T$ for columnar ice production cases.

Because processes responsible for the formation of ice particles in single-layer and multilayered clouds may be be different,
the classification of the cloud cases was performed. Using CTT and $T_{\text{column\_top}}$, we define $\Delta T$ as the temperature difference between them. The larger $\Delta T$ is, the lower inside of the observed cloud system the columns are formed. Since The relative occurrence of $\Delta T$ also shows two peaks as presented in Fig. 6 (b). Specifically, one peak is close to $\Delta T = 0$ °C, indicating that ice columns are generated close to the cloud top. The second $\Delta T$ peak is around 10 °C.


Given the distinct distribution of $\Delta T$, we have grouped the recorded clouds into the following three categories.

- Type 1: $\Delta T \leq 2\,°C$ - columnar ice production at cloud top

- Type 2: $2\,°C < \Delta T \leq 12\,°C$ - multilayered cloud

- Type 3: $12\,°C < \Delta T$ - multilayered cloud

5   Representative events of the above cloud types are presented below.

### 4.3.1   Columnar ice production at cloud top: $\Delta T \leq 2\,°C$

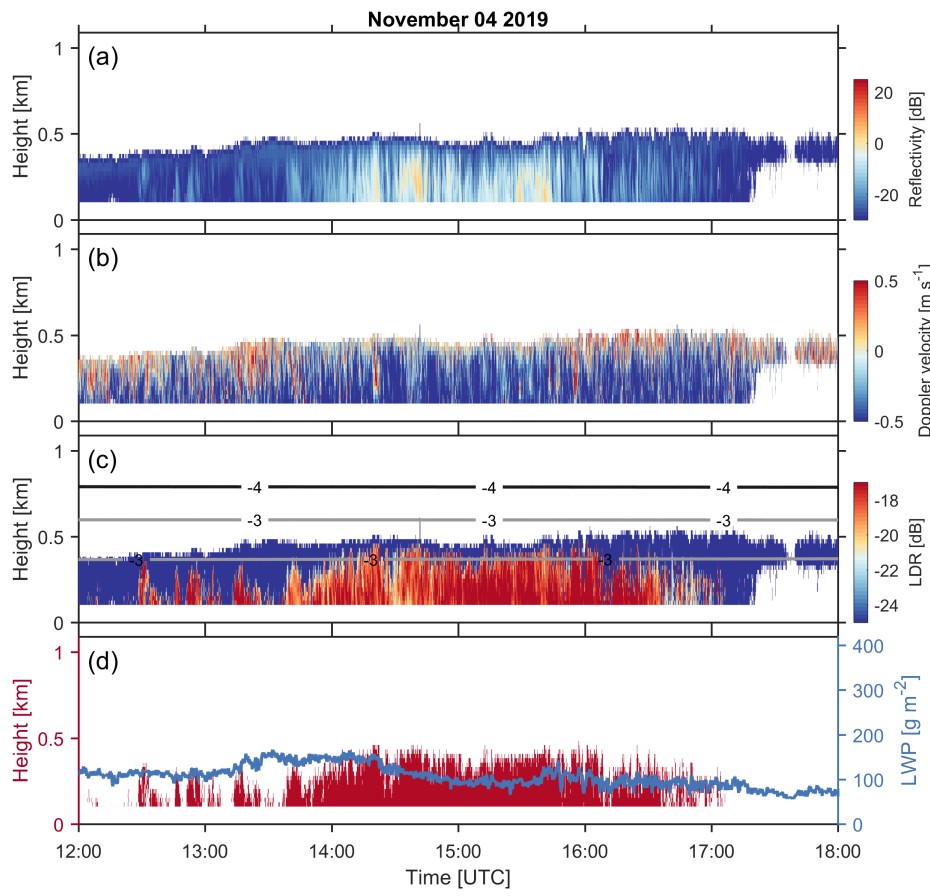

**Figure 7.** The columnar-ice-producing event on 4 November, 2019. HYDRA-W observations of (a) equivalent reflectivity factor, (b) mean Doppler velocity, where negative velocity indicates downward motion, and (c) LDR. Panel (d) presents the (left axis) detected columnar ice region and (right axis) LWP observed by HYDRA-W. The lines in (c) are isotherms produced by ICON.





This type of clouds is usually single-layer and ice columns are generated close to the cloud top. Figure 7 presents such an event on 4 November 2019. The precipitation intensity is relatively light with the CTT at around -3 °C. The W-band reflectivity close to the surface increases to around 0 dB between 14:00 UTC and 16:00 UTC. This region coincides with the enhanced LDR observations, which reaches values as high as -15 dB. Such high LDR values indicate that the dominant ice particle
type during this period is columns. As shown in Fig. 7 (b), the cloud top is turbulent and seems to be capped by an inversion layer (Fig. 7c). The observed LWP ranges between 80 and 150 $g\ m^{-2}$. This cloud with relatively low reflectivity persisted over Hyytiälä for about one day (not shown). Given the warm cloud top, the primary ice nucleation may not fully explain the significant columnar ice production (DeMott et al., 2010), as will be discussed in more detailed later. Regarding the SIP, the H-M process does not seem to be active since it requires falling ice particles serving as rimers to produce ice splinters (Hallett
and Mossop, 1974).

As shown in Fig. 6, around 22 % of columnar ice production cases are attributed to single-layer shallow clouds. Bühl et al. (2016) have also observed the prevalence of high LDR values for mixed-phase clouds with the CTT of -5 °C. They speculated that these particles are formed mainly by primary ice nucleation instead of the SIP. Recently, Yang et al. (2020) have reported a similar event of shallow stratiform clouds over the tropical ocean and found that neither INP or known SIP mechanisms can
fully explain the strong production of ice particles in such clouds where top temperatures were greater than -8 °C. In this study, we find that similar clouds also frequently occur over Hyytiälä.

### 4.3.2  Columnar ice production in multilayered clouds: $2\,°C < \Delta T \leq 12\,°C$

The event that took place on 13 February 2018 is representative of the second cloud type, as defined by $\Delta T$. As shown in Fig. 8 (a), the precipitation intensity during this event is higher than during the discussed single-layer shallow cloud case. The cloud
top temperature of the upper cloud layer is about -15 ∼ -12 °C. Before 08:00 UTC, the observed LWP is close to 100 $g\ m^{-2}$ and the mean Doppler velocity at around 0.8 km is relatively small ($\sim 1\ m\ s^{-1}$) which indicates that particles are unrimed or very lightly rimed. From 08:00 to 09:30 UTC, the falling snowflakes seem to be heavily rimed, as revealed by rather high LWP (from 200 $g\ m^{-2}$ to over 400 $g\ m^{-2}$) and mean Doppler velocity measurements ( 1.2 ∼ 1.8 $m\ s^{-1}$)(Kneifel and Moisseev, 2020). The high LWP period coincides with the region of ice columns (Fig. 8d). During this period, the observed LDR values
are enhanced, but still relatively small. This is due to masking of needle LDR signal by larger snowflakes.

This type of clouds frequently occurs over Hyytiälä (Fig. 3 and Fig. 6). In this event, the presence of supercooled liquid water may not be directly determined, however, the enhanced LWP values are indicative of the supercooled liquid water generation. In addition, the falling ice particles between 08:00 UTC and 09:30 UTC seem to be heavily rimed, as evident from mean Doppler velocity (Kneifel and Moisseev, 2020). The combination of presence of supercooled liquid water and riming, indicates that the
H-M process could be taking place and is responsible for the columnar ice production.

### 4.3.3  Deep multilayered clouds: $\Delta T > 12\,°C$

The third cloud type is not very different from the second one and represents the tail of the observed $\Delta T$ distribution as shown in Fig. 6. This type of cloud system is a deeper precipitating system with the CTT of -60 ∼ -40 °C, see Fig. 9 for an

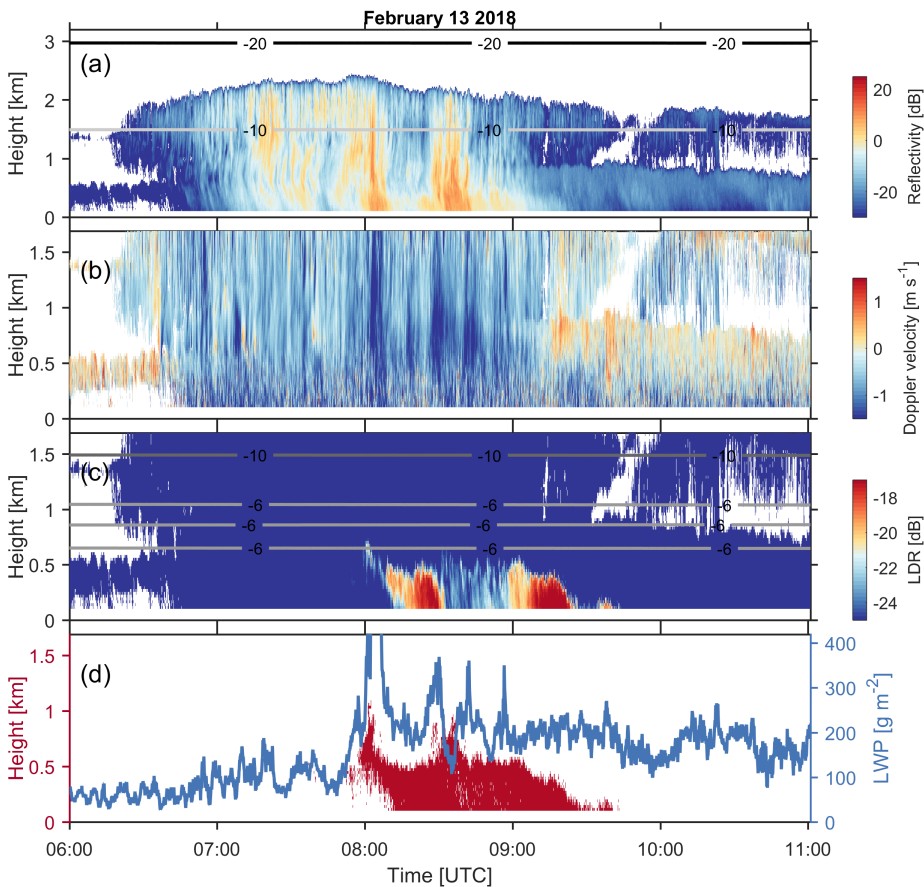

**Figure 8.** The columnar-ice-producing event recorded on 13 February, 2018. HYDRA-W observations of (a) equivalent reflectivity factor, (b) mean Doppler velocity, where negative velocity indicates downward motion, and (c) LDR. Panel (d) presents the (left axis) detected columnar ice region and (right axis) LWP observed by HYDRA-W. The lines in (a) and (c) are isotherms produced by ICON. Note that the y-axis scale in (a) is different from that in (b), (c) and (d).

example . The presented case took place on 2 February 2018. There are several features that are worthwhile to point out. The mean Doppler velocity observations exhibit signatures of atmospheric waves. Between 0500 and 0600 UTC such waves can be observed at around 1 km altitude. At the later time, the wave signatures appears at 0.5 km agl. The strongest velocity variation, observed around 06:00 UTC, seem to coincide with the LWP peak. At the same time as the waves are observed signatures of columnar ice production are also detected, pointing to a possible connection between the two.

5  Deep precipitating clouds usually have large number of ice crystals formed at cloud top. Given the large ice flux, manifested by the higher radar reflectivity values, in this precipitation system, it is difficult for supercooled liquid droplets to survive. The supercooled water droplets can be rapidly depleted through the Wegener–Bergeron–Findeisen (WBF) process (Korolev et al.,

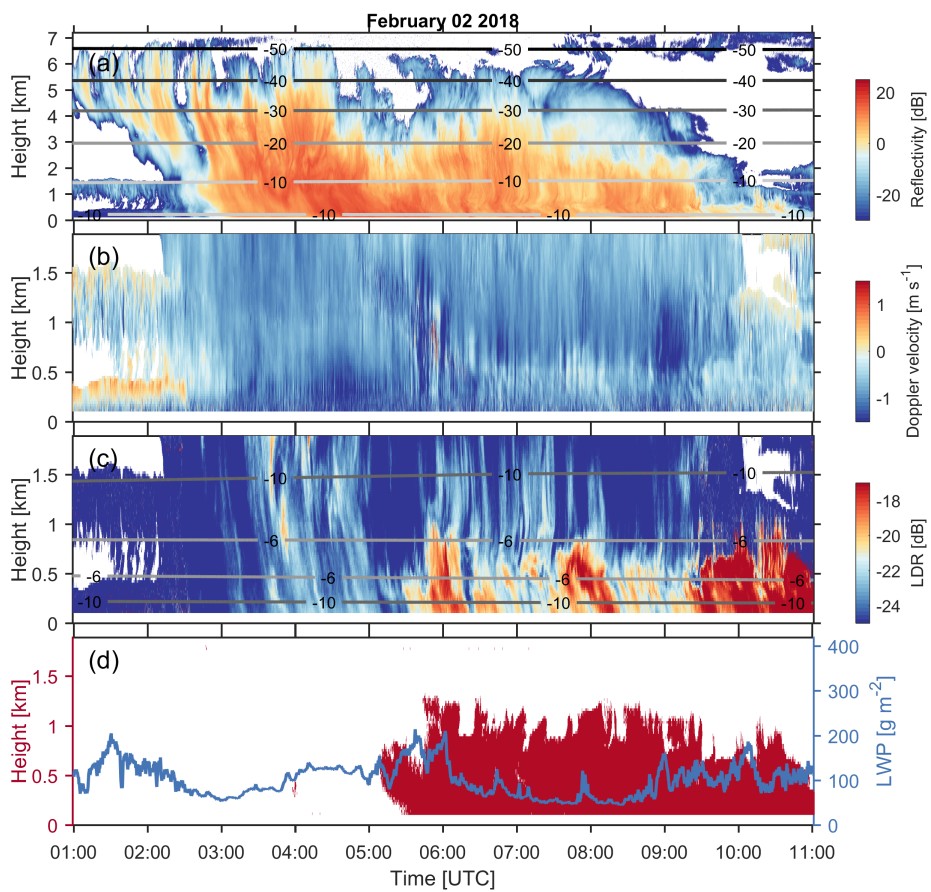

**Figure 9.** Same as Fig. 8, but for the columnar-ice-producing event observed on 2 February 2018.

2017), as well as riming (Fukuta and Takahashi, 1999). Nevertheless, the atmospheric waves could generate conditions needed for forming and maintaining presence of supercooled liquid water droplets (Korolev, 1995; Korolev and Field, 2008; Majewski and French, 2020). In such cases, ice needles may be generated by the H-M process (Hogan et al., 2002; Houser and Bluestein, 2011).

5  ## 4.4  Characteristics of different columnar-ice-producing cloud types

As discussed above, we have identified three types of cloud systems where columnar ice particles form. To better understand how columnar ice production is related to cloud properties, persistence of columnar ice in these clouds and the amount of LWP were considered for further analysis.





### 4.4.1 Persistence of columnar ice crystals

As was demonstrated by the case studies, the columnar ice production may persist over several hours and therefore these particles may play a major role in determining cloud properties. To document this, we have derived statistics of the columnar ice production persistence for each cloud case. This was done by computing duration of a continuous columnar ice production event. Since in some cases, for example in the presented shallow single-layer clouds (Fig. 7) formation of columns may be intermittent in nature (as can also be seen in Luke et al., 2021), similar to (Shupe et al., 2011) gaps of less than 30 min were accepted. In addition, cases persisting less than 1 min were removed. As shown in Fig. 10, $40 \sim 50\ \%$ of columnar-ice-producing events persist for less than 1 hour. However, there is a significant fraction that could last for more than 3 or even 6 hours. This hints that the production of ice columns play an important role in defining cloud properties and should be included while considering radiative or precipitation properties of such clouds Fig. 9).

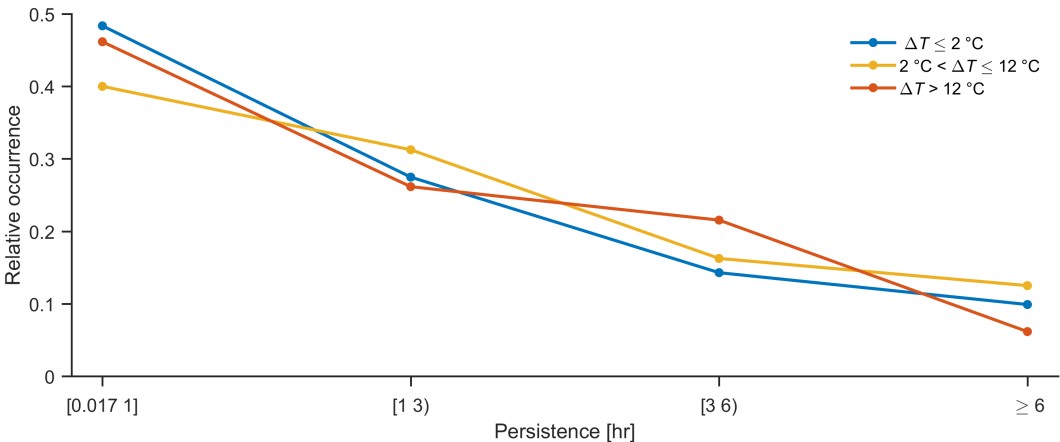

**Figure 10.** Relative occurrence of the persistence of columnar-ice-producing events from February 2018 to April 2020.

### 4.4.2 LWP

Presence of liquid water droplets may be a necessary condition for the formation of the observed ice columns. Compared to clouds where no ice columns were detected, columnar-ice-producing clouds have higher LWP values (Fig. 4). For different columnar-ice-producing cloud types, the observed LWP values seem to be somewhat different. Figure 11 shows LWP occurrence for these three cloud types. In general, their distributions are similar, while identifiable differences still exist. The median value of LWP for single-layer clouds is the lowest and the second type of clouds has the highest LWP. Interestingly, while both the second and third types of clouds are multilayered, the LWP of the second type is delectably higher than for the third one. Comparing Figs. 8 and 9, we find that the radar reflectivity above the columnar ice layer is generally higher in the third cloud type than the second one. The INP concentrations in clouds with a cold top are expected to be higher than for warmer clouds, hence more falling ice particles are expected for deep precipitating clouds. Therefore, we speculate that one of the reasons




responsible for the difference in LWP may be the ice number concentration, which is related to the consumption of the liquid water via the WBF process (Lamb and Verlinde, 2011) and riming (Fukuta and Takahashi, 1999).

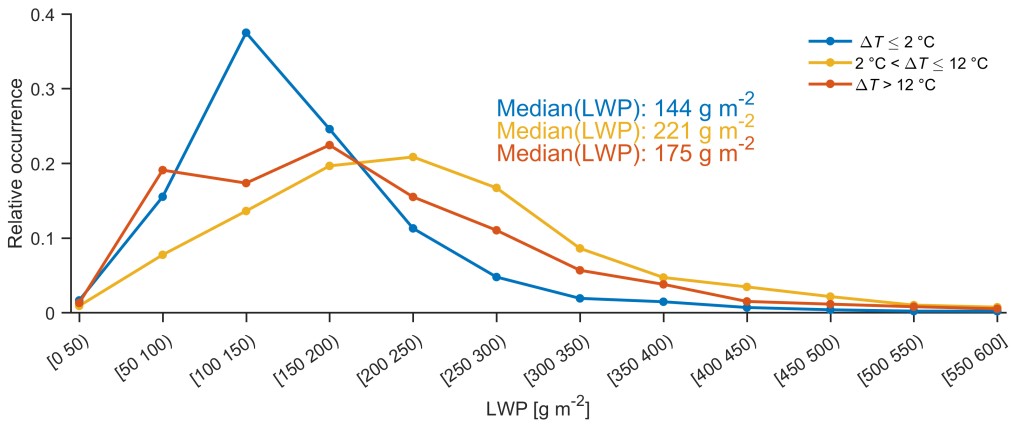

**Figure 11.** Relative occurrence of LWP for all columnar-ice-producing events.

## 5 Potential role of SIP in columnar ice production in single-layer shallow clouds

Given the rather frequent formation of ice crystals at temperatures warmer than - 10 C, where expected INP concentrations are low, it is important to investigate a potential role of SIP. In multilayered clouds, identified here as cloud types II and III, the ice formation can be enhanced by the H-M process (e.g., Grazioli et al., 2015; Giangrande et al., 2016; Sinclair et al., 2016; Keppas et al., 2017; Sullivan et al., 2018; Gehring et al., 2020). In single-layer clouds the mechanisms for ice multiplication are less investigated. Recent studies by Lauber et al. (2018); Keinert et al. (2020); Lauber et al. (2021) have shown that freezing fragmentation may play such a role. To study if these columnar ice particles can be attributed to the SIP, the estimated ice number concentrations are compared to the expected INP concentrations. If the derived number concentrations of ice exceed those of INPs, we can conclude that the SIP is potentially active in such clouds. The concentration of INP was computed by using CTT using different parameterizations reported in literature, i.e.

- Fletcher (1962) parameterization based on INP measurements obtained below -10 °C.

- Cooper (1986) parameterization which is not directly derived from INP measurements but the observed ice number concentrations when the impact of SIP is minimized. The temperature of measurements is between -30 °C and -5 °C.

- DeMott et al. (2010) parameterization based on INP measurements from 9 sites between -35 °C and -9 °C. In our study we have used the average INP concentration - temperature relation presented in DeMott et al. (2010).

- Schneider et al. (2020) parameterization derived from the INP measurements obtained at Hyytiälä. The temperature range is -20 $\sim$ -8 °C. Also from this study the average INP concentration - temperature relation was used.



As was previously discussed, the INP parameterizations differ significantly at this temperatures (-10 ∼ 0 °C), as shown in Fig. 12. It should be noted that not all parameterizations were derived using observations at these cloud temperatures and some of them were extrapolated beyond their validity range. The most interesting comparison is to (Schneider et al., 2020), which is based on observations at Hyytiälä collected during 2018, so their observation period at least partially overlaps with ours. It should also be pointed out that INP observations were carried out at the ground, where INP concentrations are typically higher.

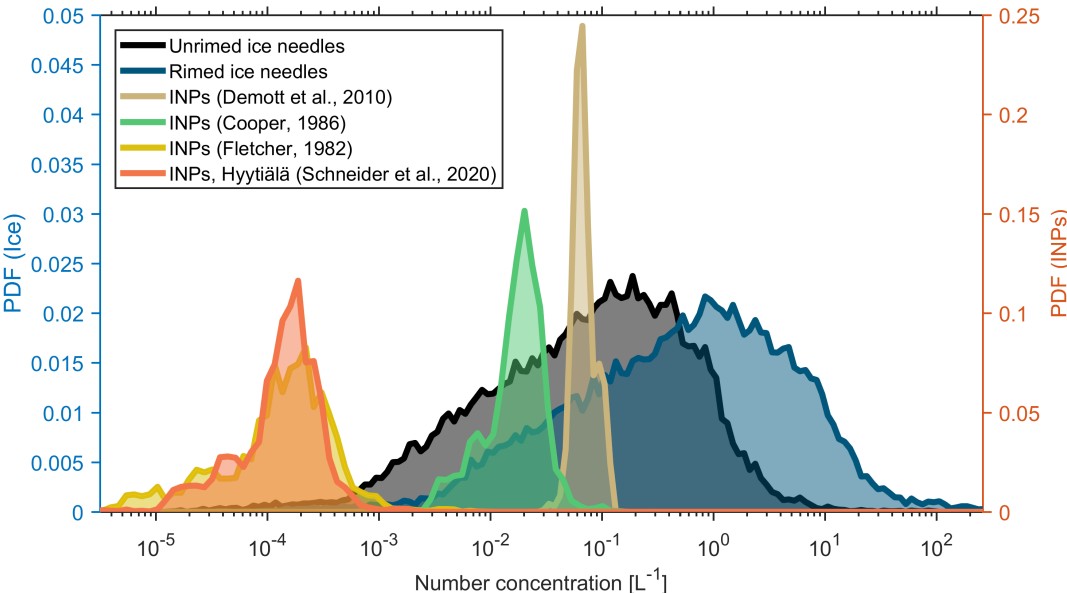

**Figure 12.** Probability density function (PDF) of estimated columnar ice (left) and INP (right) concentrations in shallow single-layer clouds.

Radar-based retrieval of particle number concentration is rather uncertain. Because of this uncertainty, the derived number concentrations should be treated as our best estimates of the order of magnitude of the ice column number concentration. As shown in Fig. 4, the observed LWP for columnar-ice-producing clouds is significantly higher than those without ice columns. Hence, pristine ice crystals are anticipated to grow in mixed-phase conditions and ice needles rather than solid columns are expected to form (Lamb and Verlinde, 2011). This limits the parameter space, where we need to search for microphysical properties of ice particles to constrain our retrieval.

The retrieval is based on estimating ice water content from radar observations, following (Hogan et al., 2006), as:

$$\mathrm{IWC} = 0.00058ZT + 0.0923Z - 0.00706T - 0.992 \,[g\,m^{-3}], \tag{1}$$

where $Z$ is the W-band radar reflectivity and T is the air temperature. Because, in a single-layer shallow cloud ($\Delta T \leq 2$ °C), ice needles are the predominant precipitation particles, radar reflectivity measured close to the ground, in the fourth radar gate, 329 m amsl or 179 m agl, is used in this retrieval. The selection of the reflectivity measured close to the ground helps to limit potential attenuation problems, as well. Using IWC the number concentration of ice needles, $N_{\mathrm{needle}}$ , can be estimated as





follows:

$$N_{\text{needle}} = \frac{\text{IWC}}{m_{\text{needle}}} \tag{2}$$

where $m_{\text{needle}}$ is mass of a characteristic ice needle. The introduced uncertainty at this step depends on the definition of the characteristic needle. Here, we use mean Doppler velocity (MDV) and velocity-mass relation to estimate $m_{\text{needle}}$. Since MDV

is reflectivity weighted, $m_{\text{needle}}$ would be mainly determined by larger ice particles, and therefore the resulting $N_{\text{needle}}$ is underestimated. For the purpose of this study, this underestimation is not a major issue, because we want to test if the observed $N_{\text{needle}}$ exceeds expected INP concentrations.

There are a number of reported ice needle properties. To take into account potential differences in ice needle properties, two relations linking velocity and mass by Kajikawa (1976) for rimed needles, and (Heymsfield, 1972) for unrimed needles were

used. For rimed ice needles, the relation between terminal fall velocity and mass was derived by Kajikawa (1976) and can be written as:

$$v_{\text{needle, rimed}} = 1.55 m_{\text{needle, rimed}}^{0.271} \left[ \frac{\rho(z)}{\rho(1024 \text{ m})} \right]^{0.4} [m \ s^{-1}] \tag{3}$$

where $m_{\text{needle, rimed}}$ is the mass of rimed ice needles, and the term $\left[ \frac{\rho(z)}{\rho(1024 \text{ m})} \right]^{0.4}$ accounts for the change in air density $\rho$ at a given height $z$ (Heymsfield et al., 2007). In our study z is 329 m and in Kajikawa (1976) the altitude where needles were

observed is 1024 m. For unrimed needles, Heymsfield (1972) has derived a relation linking terminal fall velocity and needle length, $L$. The terminal velocity of unrimed needles at a given height can be estimated as follows (Heymsfield, 1972):

$$v_{\text{needle, unrimed}} = \left( 0.0006 + 0.2796L - 0.0497L^2 + 0.0041L^3 \right) \left[ \frac{\rho(z)}{\rho(0 \text{ m})} \right]^{0.4} [m \ s^{-1}] \tag{4}$$

The needle can be modeled as a cylinder, the mass of which is:

$$m_{\text{needle, unrimed}} = \frac{\pi}{4} \rho_{\text{ice}} L D^2 \tag{5}$$

where $\rho_{\text{ice}}$ [$g \ cm^{-3}$] and $D$ [mm] denote density of unrimed needles and the minor axis, respectively. Their parameterizations have been given by Heymsfield (1972):

$$\rho_{\text{ice}} = 0.6 L^{-0.117} \tag{6}$$

and

$$D = 0.1973 L^{0.414}. \tag{7}$$

Applying the power law fit to $m_{\text{needle, unrimed}}$ and $v_{\text{needle, unrimed}}$ values when $L \in [0.03 \ 5]$ [mm] yields:

$$v_{\text{needle, unrimed}} = 1.09 m_{\text{needle, unrimed}}^{0.377} \left[ \frac{\rho(z)}{\rho(0 \text{ m})} \right]^{0.4}. \tag{8}$$





The observed MDV is affected by vertical air motion. To at least partially mitigate this issue, the observed MDV is averaged over 20 min (Protat and Williams, 2011; Mosimann, 1995; Kneifel and Moisseev, 2020; Silber et al., 2020). While this step reduces impact of air motion by averaging Doppler velocity over updrafts and downdrafts, the residual air motion is expected to widen the retrieved distribution of $N_{\mathrm{needle}}$.

The derived number concentrations of ice particles are compared to expected INP concentrations (Fig. 12). The results show that the estimated ice number concentrations for the rimed needles (Kajikawa, 1976) is generally larger than that of unrimed (Heymsfield, 1972). Regardless of the difference between rimed and unrimed needles, and INP parameterization used, there seems to be a large fraction of the cases where INP concentrations are not sufficient to explain observed $N_{\mathrm{needle}}$. Our results are similar to the conclusion reached by (Luke et al., 2021), who have used a different approach for the establishing the range of
ice crystal concentration from radar observations. For many cases $N_{\mathrm{needle}}$ is in the order of $10^{-2} \sim 10^{1}\ L^{-1}$, which is similar with the number observed in tropical stratiform clouds (Yang et al., 2020).

The significant discrepancy between INP concentrations estimated from INP parameterizations and retrieved ice number concentrations indicates that primary ice nucleation does not seem to be the only mechanism responsible for the formation of ice particles in these shallow clouds. Because, the analysis was performed on shallow single-layer clouds, this discrepancy may
not be explained by the H-M process (Hallett and Mossop, 1974) since no rimers are falling from upper clouds. So it appears that other, less studied, SIP mechanisms may play an important role in amplifying ice number concentrations in such shallow clouds. This conclusion is inline with a number of other studies. For example, Knight (2012) have found that the SIP may take place at -5 ° at the absence of rimers, for which the cause is still under investigation. Recently, similar finding reported for stratiform clouds over the tropical ocean by Yang et al. (2020). They have speculated that droplet collisional freezing (Hobbs,
1965; Alkezweeny, 1969) and pre-activated INPs (Roberts and Hallett, 1968; Mossop, 1970) could be responsible for this discrepancy. Recent laboratory studies (Lauber et al., 2018; Keinert et al., 2020) have shown that freezing breakup may be a source of secondary ice particles. Luke et al. (2021) have also suggested that freezing breakup may be more efficient than the H-M process in nature.

## 6   Conclusions

This study documents formation of ice particles in clouds at the temperatures of -10 °C or higher. The analysis was performed using W-band cloud radar observations collected at the University of Helsinki Hyytiälä station from February 2018 to April 2020. The columnar ice particles were identified using measurements of spectral LDR. It was found that ice particle formation is relatively frequent in such clouds. The radar observations detect formation of ice particles in $5 \sim 13$ % of clouds, where cloud top temperature is -12 °C or warmer. In colder clouds this percentage can be as high as 33 %. The columnar ice producing
clouds tend to have higher LWP, potentially indicating that supercooled water droplets are important for formation of the observed ice particles. It was also observed that columnar ice production seems to have a significant impact on the surface precipitation. This effect is especially important for warmer clouds.



Using the temperature difference, $\Delta T$, between the altitudes where columns are first detected and cloud top, the columnar-ice-producing clouds were subdivided into three categories. First category, where $\Delta T$ is less or equal to 2 °C , represents shallow single-layer clouds. In these clouds ice particles are forming at or close to the cloud top. The other two categories, where $2\,°C < \Delta T \leq 12\,°C$ and $\Delta T > 12\,°C$ , represent deeper multi-layered clouds. In multi-layered cloud systems columnar

ice crystals are forming in lower cloud layer seeded by ice particles falling from upper cloud levels. It was observed that 40 $\sim$ 50 % of columnar ice production cases persist for 1 hour or less, while in some cases they can persist for over 6 hours. The distributions of LWP values for the three types of columnar-ice-producing clouds are somewhat different. The median LWP value is the largest ($221\ g\ m^{-2}$) in clouds where $2\,°C < \Delta T \leq 12\,°C$. Such high LWP could favor riming and cause Hallet-Mossop process. To make a definite conclusion, however, a more thorough study, where locations of supercooled liquid

layers is identified, is needed. For the single-layer shallow clouds, number concentrations of ice columns were derived from the radar observations. It was observed that the concentration of ice particles exceeds expected concentration of INP for a large number of cases. This indicates that a SIP mechanism is active in these clouds. Given that in single-layer shallow clouds, there are no rimers that could cause H-M process, we advocate that another SIP process may play a role here.

*Data availability.* W-band radar data is available from https://doi.org/10.23729/5aab6b78-90c9-49ab-8264-f4168528a0f3. ICON data is

available from http://devcloudnet.fmi.fi/.

*Author contributions.* DM conceptualized the study. HL performed the investigation and wrote the draft. OM, PT and DM contributed to reviewing and editing this draft.

*Competing interests.* The authors declare that they have no conflict of interest.

*Acknowledgements.* We would like to thank the personnel of Hyytiälä station for their support in field observation. We want to acknowl-

edge Matti Leskinen for his work in radar maintenance. This research has been supported by the Horizon 2020 (grant nos. ACTRIS-2 654109, ACTRIS PPP 739530, ACTRIS-IMP 871115, ATMO-ACCESS 101008004, ERA-PLANET iCUPE, 689443) and Academy of Finland (ACTRIS-NF 328616, ACTRIS-CF 329274, NanoBioMass 307537, ACRoBEAR 334792, and Center of Excellence in Atmospheric Sciences, 307331, Atmosphere and Climate Competence Center, 337549), University of Helsinki (ACTRIS-HY). Haoran Li was also funded by China Scholarship Council.

We acknowledge the European Research Infrastructure for the observation of Aerosol, Clouds and Trace Gases (ACTRIS) for providing the DWD ICON (Global) model data, which was produced by the Deutscher Wetterdienst (DWD) and the Finnish Meteorological Institute, and is available for download from https://cloudnet.fmi.fi/.





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
