# Peer review of "Two-year statistics of columnar ice production in stratiform clouds over Hyytiälä, Finland: environmental conditions and the relevance to secondary ice production"

_Atmospheric Chemistry and Physics, 2021_

## Referee Comment (RC1)

Atmospheric Chemistry and Physics - Manuscript acp-2021-332:
"Multiyear statistics of columnar ice production in stratiform clouds over Hyytiälä, Finland"
by H. Li, O. Möhler, T. Petäjä and D. Moisseev

**1  Summary**

This manuscript investigates the formation of columnar ice crystals in different types of (winter) clouds at temperature warmer than $-10\,°C$, by analyzing two years of vertical profiles of reflectivity, Doppler velocity and LDR collected by a W-band polarimetrioc radar over the Hyytiälä site in Finland. Environmental conditions (mostly temperature and relative humidity) are provided by the ICON NWP model. The identification of columnar ice crystals is based on their typical linear depolarization ratio (LDR) signature, and their occurrence can be quantified and analyzed as a function of the cloud type (single vs multi-layer cloud) as well as the temperature difference between cloud top and the altitude of columnar ice formation. Those analyses suggest that (i) columnar ice formation is relatively frequent (from 5 to 30% depending on the cloud top temperature), (ii) columnar ice formation is associated with more intense surface precipitation, (iii) the liquid water path (LWP) is larger when columnar ice crystals are produced compared to when there is no such production, and (iv) some secondary ice production (SIP) mechanisms are likely involved to explain the larger number of ice crystals than ice nucleating particles in the case of single-layer clouds.

**2  Recommendation**

Taking advantage of a nice set of vertical profiles of radar observations at W-band over well-known site in Finland, this work relevantly combines radar measurements and model output to investigate the importance (in terms of occurrence and impact on surface precipitation) of columnar ice production in "warm" ice or mixed-phase clouds and to characterize those. In addition, the discrepancy found between the estimated order of magnitude of ice crystal number on the one hand and of ice nucleating particle number on the other, the latter being smaller, suggest that secondary ice production mechanisms must be active in order to complement primary ice production. The proposed approach and the obtained results are sound (although some aspects remain rather speculative), and they are relevant for the ACP readership. I do not have major concerns, so I recommend to send the manuscript back to the authors for minor revisions. A list of comments and questions is provided below.

**3  Specific comments**

1. P.2, l.2-3: in high latitude regions (Arctic, Antarctic, Southern Ocean), INP concentration has been estimated to be lower than at mid-latitudes (e.g. *DeMott et al.*, 2016; *Wex et al.*, 2019).

2. P.2, l.18: other SIP processes than HM have been already implemented in different atmospheric models. For instance: *Hoarau et al.* (2018); *Sullivan et al.* (2018); *Zhao and Liu* (2021).

3. P.4, l.9-13: what is the horizontal resolution of ICOM simulations above Hyytälä?

4. P.6, l.24: the used statistics of relative humidity from the ICOM simulations likely depend on some of the microphysica parameterizations. This should be mentioned and those parameterizations could be listed (no need of exhaustive descriptions).

5. P.8, l.12-13: "the majority of columnar ice production cases took in areas of high supersaturation, which is potentially favorable for liquid water droplet formation or existence", according to Fig.2.b, most cases are below 100% of RH with respect to liquid water...

6. P.8, l.17: smaller than $-8\,°C$ would be more correct I think.

7. P.10, l.8: the argument that "precipitation processes are more complex" is a bit short, please elaborate.

8. P.12, l.11-16: I would suggest to make a more explicit reference to Section 5 here, to inform the reader that this question is addressed later in Section 5.

9. P.12, l.30: "is responsible" should be "could be responsible" as this is speculative.

10. P.12, Section 4.3.2: there is no reason (even speculative) provided to explain how the supercooled liwuid water could be generated in such clouds...

11. P.13, l.2-3: it was not totally clear to me where are these waves in Fig.8.b, maybe circling them would help...

12. P.13, l.2-5: more explanations about those waves (nature, origin...) shold be provided. If they are related to larger scale processes, are they reproduced in ICON simulations?

13. P.13, l.5: "pointing to a possible connection between the two": between 06:30 and 07:30, I do not see any wave signature (I may have missed it) but the period is still identified as "columnar ice productive", so the connection is not that clear in my view...

14. P.14, l.3: the explanation about the atmospheric waves generating SLW is too short (and speculative): the Doppler velocities (Fig.9.b) do not show significant updraft around 1 km of altitude between 06:00 and 10:00 while columnnar ice crystals are identified by their LDR signature...

15. P.15, l.17: detectably?

16. P.16, l.6-9: there are also ice only SIP mechanisms (e.g. *Korolev and Leisner*, 2020).

17. P.18.l.25: a " ," is missing between 0.03 and 5.

18. P.19, l.27-28: "It was found that... formation of ice particles": I find this formulation rather confusing: this work focuses on the formation of columnar ice crystals at temperatures warmer than $-10\,°C$, not on the formation of ice crystals in general.

**References**

DeMott, P. J., et al., Sea spray aerosol as a unique source of ice nucleating particles, *Proc. Natl. Acad. Sci. USA*, *113*(21), 5797–5803, doi:10.1073/pnas.1514034112, 2016.

Hoarau, T., J.-P. Pinty, and C. Barthe, A representation of the collisional ice break-up process in the two-moment microphysics lima v1. 0 scheme of meso-nh, *GMD*, *11*(10), 4269–4289, doi:10.5194/gmd-11-4269-2018, 2018.

Korolev, A., and T. Leisner, Review of experimental studies of secondary ice production, *Atmos. Chem. Phys.*, *20*(20), 11,767–11,797, doi:10.5194/acp-20-11767-2020, 2020.

Sullivan, S. C., C. Barthlott, J. Crosier, I. Zhukov, A. Nenes, and C. Hoose, The effect of secondary ice production parameterization on the simulation of a cold frontal rainband, *ACP*, *18*(22), 16,461–16,480, doi:10.5194/acp-18-16461-2018, 2018.

Wex, H., W. Huang, L. Zhang, H. Hung, R. Traversi, S. Becagli, R. J. Sheesley, C. E. Moffett, T. E. Barrett, and R. Bossi, Annual variability of ice-nucleating particle concentrations at different arctic locations, *Atmos. Chem. Phys.*, *19*(7), 5293–5311, doi:10.5194/acp-19-5293-2019, 2019.

Zhao, X., and X. Liu, Global importance of secondary ice production, *GRL*, *48*(11), e2021GL092,581, doi:10.1029/2021GL092581, 2021.

---

## Referee Comment (RC2)

The paper is well written and contains nice results in terms of ice properties in mixed phase clouds and ice production derived from cloud radar observations. I have a small list of suggestions to improve the paper.

Only minor revisions are required.

Title: Two-year statistics… is more appropriate… (Feb 2018 to April 2020)

P1, l2: remote sensing IS an observation. So, remove 'observation' here, please.

P1, l9: that that

P1, l15: Please use 2 K (and always Kelvin for the temperature difference throughout the paper)

P2, l1: provide a reference please

P2, l3: I am a bit confused…, dust particles (mineral dust) are about three orders of magnitude better INPs than marine particles (sea salt) at the same temperature. Please check, McCluskey (JGR 2018, Fig 8 (marine) vs Niemand 2012 (dust) similar to DeMott, 2015, dust).

P3, l31: What is the LDR decoupling of the system? It would be interesting what is the minimum and maximum LDR that can be detected within these clouds for the usual ranges of Z, Signal and LDR.

P4, l10: The website is depreciated, now it is cloudnet.fmif.fi .
Further remark: For ACTRIS/Cloudnet datasets DOIs are available. Please refer to the specific dataset that was used in this study with such a DOI.

P4, l30: Here, the Myagkov (2016) references would be appropriate to be included:
Myagkov, A., Seifert, P., Bauer-Pfundstein, M., and Wandinger, U.: Cloud radar with hybrid mode towards estimation of shape and orientation of ice crystals, Atmos. Meas. Tech., 9, 469–489, https://doi.org/10.5194/amt-9-469-2016, 2016.
Myagkov, A., Seifert, P., Wandinger, U., Bühl, J., and Engelmann, R.: Relationship between temperature and apparent shape of pristine ice crystals derived from polarimetric cloud radar observations during the ACCEPT campaign, Atmos. Meas. Tech., 9, 3739–3754, https://doi.org/10.5194/amt-9-3739-2016, 2016.

P5, l3: Again a reference to Radenz et al. (2019) would be appropriate here.

P5, Figure 1 (c), this third peak (the last one to the right) indicates the presence of liquid water. Could the columnar ice particles actually have been produced by primary ice formation in this liquid layer and not be a product of ice multiplication?

P7, Figure 2(b): These statistics are highly questionable. What do they show us? Is ICON able to derive actual realistic humidity values for the clouds under study? Did ICON even resolve the clouds under study?

P10, Figure 6(a): The figure seems to contain mixed-phase and ice clouds. Would it be possible to show both species in this figure separately?

P10, l18: Again, using unit Kelvin (…10 K) is really important here to differentiate between temperature differences and absolute temperature values.

P11, Figure 7 (d): Are the LWP values reliable if precipitation (especially rain) reaches the ground? It is known that microwave radiometers have a problem with such conditions - is that also true for the radar-derived LWP?

P16, Section 5: This chapter is rather speculative and does not show a conclusive result.
- $N_{neelde}$ should be computed taking into account that the size of needles is distributed spectrally. (I think the term "characteristic needle" points to this fact.)
- An error analysis is necessary here. What is the impact of residual measurement errors in fall velocity on the ice crystal number concentration?
- INP is compared against ice number concentration: A thorough error analysis for both values is needed.
- How is ambient INP concentration over the measurement site derived?

P19, l10: The last sentence of the paragraph is not clear, please explain in more detail. "For many cases" is a bit blurry.

---

## Author Comment (AC1)

**Reviewer # 2**

1 Summary

This manuscript investigates the formation of columnar ice crystals in different types of (winter) clouds at temperature warmer than −10 ∘C, by analyzing two years of vertical profiles of reflectivity, Doppler velocity and LDR collected by a W-band polarimetrioc radar over the Hyytiala site in Finland. Environmental conditions (mostly temperature and relative humidity) are provided by the ICON NWP model. The identification of columnar ice crystals is based on their typical linear depolarization ratio (LDR) signature, and their occurrence can be quantified and analyzed as a function of the cloud type (single vs multi-layer cloud) as well as the temperature difference between cloud top and the altitude of columnar ice formation. Those analyses suggest that (i) columnar ice formation is relatively frequent (from 5 to 30% depending on the cloud top temperature), (ii) columnar ice formation is associated with more intense surface precipitation, (iii) the liquid water path (LWP) is larger when columnar ice crystals are produced compared to when there is no such production, and (iv) some secondary ice production (SIP) mechanisms are likely involved to explain the larger number of ice crystals than ice nucleating particles in the case of single-layer clouds.

2 Recommendation

Taking advantage of a nice set of vertical profiles of radar observations at W-band over wellknown site in Finland, this work relevantly combines radar measurements and model output to investigate the importance (in terms of occurrence and impact on surface precipitation) of columnar ice production in "warm" ice or mixed-phase clouds and to characterize those. In addition, the discrepancy found between the estimated order of magnitude of ice crystal number on the one hand and of ice nucleating particle number on the other, the latter being smaller, suggest that secondary ice production mechanisms must be active in order to complement primary ice production. The proposed approach and the obtained results are sound (although some aspects remain rather speculative), and they are relevant for the ACP readership. I do not have major concerns, so I recommend to send the manuscript back to the authors for minor revisions. A list of comments and questions is provided below.

*We sincerely appreciate the reviewer for very thorough comments on our paper. We have amended the manuscript as suggested. Please see below our response to your comments.*

3 Specific comments

1. P.2, l.2-3: in high latitude regions (Arctic, Antarctic, Southern Ocean), INP concentration has been estimated to be lower than at mid-latitudes (e.g. DeMott et al., 2016; Wex et al., 2019).

*Thank you for the nice supplement. We have added a sentence as follows:*

*In addition, it has been found that INP concentrations in high latitudes are generally lower than in mid-latitudes (e.g., Demott et al., 2016, Wex et al., 2019).*

2. P.2, l.18: other SIP processes than HM have been already implemented in different atmospheric models. For instance: Hoarau et al. (2018); Sullivan et al. (2018); Zhao and Liu (2021).

*Agree. We have added these works at proper places, and have amended the previous statement as:*

*This is referred as to Hallett–Mossop (H-M) process, the most studied and most frequently implemented SIP mechanism in numerical models (Field et al., 2017).*

3. P.4, l.9-13: what is the horizontal resolution of ICOM simulations above Hyytiälä?

*It is 13 km. In the revised manuscript, we have added such description in Section 2.2.*

4. P.6, l.24: the used statistics of relative humidity from the ICOM simulations likely depend on some of the microphysica parameterizations. This should be mentioned and those parameterizations could be listed (no need of exhaustive descriptions).

*Agree. Given $RH_{liquid}$ was calculated from $RH_{liquid}$, we have removed $RH_{ice}$ in the revised manuscript for avoiding misleading. We have added such sentence:*

*However, the values of $RH_{liquid}$ and $RH_{ice}$ should be interpreted with caution. ICON applies a liquid saturation adjustment, limiting the liquid supersaturation to saturation. $RH_{liquid}$ values exceeding 100 % are attributed to numerical artifacts. $RH_{ice}$ was calculated based on the forecasted temperature, pressure as well as $RH_{liquid}$, therefore can be affected by numerical artifacts as well. Given the uncertainty of ICON forecasts, we regard the presented statistics in Fig.2 as a sanity check for our method.*

5. P.8, l.12-13: "the majority of columnar ice production cases took in areas of high supersaturation, which is potentially favorable for liquid water droplet formation or existence", according to Fig.2.b, most cases are below 100% of RH with respect to liquid water...

*Thank you for pointing this out. This sentence has been amended as*

*the majority of columnar ice production cases took in areas close to liquid supersaturation.*

6. P.8, l.17: smaller than −8 ◦C would be more correct I think.

*Amended.*

7. P.10, l.8: the argument that "precipitation processes are more complex" is a bit short, please elaborate.

*We have amended this sentence as*

*Colder clouds are prone to be consisting of the multiple cloud layers where precipitation processes are affected by multiple processes, such as riming, aggregation, sublimation, at various levels (Houze Jr and Medina, 2005, Verlinde et al., 2013, Moisseev et al., 2015).*

8. P.12, l.11-16: I would suggest to make a more explicit reference to Section 5 here, to inform the reader that this question is addressed later in Section 5.

*Thank you for the constructive suggestion.*

*In this study, we find that such clouds also frequently occur over Hyytiälä, and more detailed analysis will be presented in Sec. 5.*

9. P.12, l.30: "is responsible" should be "could be responsible" as this is speculative.

*Amended*

10. P.12, Section 4.3.2: there is no reason (even speculative) provided to explain how the supercooled liquid water could be generated in such clouds...

*Due to the lack of sounding observations, it is challenging to identify the mechanism of generating supercooled liquid water. Therefore, we tend to be conservative on explain the presence of liquid. Also, we did find such clouds resemble with some previous studies, and we will do further analysis in the future. We have added the discussions below:*

*Although sounding measurements were absent, this cloud type seems to be very similar with the one reported by Westbrook and Illingworth (2013), namely, a layer of supercooled liquid water with*

*the top temperature of around -15℃ seeding low-level stratus clouds in the boundary layer.*

11. P.13, l.2-3: it was not totally clear to me where are these waves in Fig.8.b, maybe circling them would help...

*We have added the zoom-in view on the waves in Fig. 8b.*

12. P.13, l.2-5: more explanations about those waves (nature, origin...) should be provided. If they are related to larger scale processes, are they reproduced in ICON simulations?

We agree that these wave signatures are interesting. Our separate study (Li et al., ACP, 2021), with thorough analysis, has reported that Kelvin-Helmholtz waves with such velocity oscillations are favorable for large liquid drops formation and SIP. We are currently looking in-depth on such phenomenon, and more detailed explanations will come in our future studies.

13. P.13, l.5: "pointing to a possible connection between the two": between 06:30 and 07:30, I do not see any wave signature (I may have missed it) but the period is still identified as" columnar ice productive", so the connection is not that clear in my view...

We have added the zoom-in view on the waves in Fig. 8b. The connection should be more visible.

14. P.14, l.3: the explanation about the atmospheric waves generating SLW is too short (and speculative): the Doppler velocities (Fig.9.b) do not show significant updraft around 1 km of altitude between 06:00 and 10:00 while columnnar ice crystals are identified by their LDR signature...

We agree with the reviewer. We have added the zoom-in view on the waves in Fig. 8b. The link between waves and SLW, SIP is exactly a topic that we are investigating and we have discussed it in a separate study (Li et al., ACP, 2021). We have referred to (Li et al., ACP, 2021) in the revised manuscript.

15. P.15, l.17: detectably?

Corrected.

16. P.16, l.6-9: there are also ice only SIP mechanisms (e.g. Korolev and Leisner, 2020).

Agree. We have amended this sentence as

*In multilayered clouds, identified here as cloud types II and III, it has been found that the ice formation can be enhanced by the H-M process (e.g., Grazioli et al., 2015; Giangrande et al., 2016; Sinclair et al., 2016; Keppas et al., 2017; Sullivan et al., 2018; Gehring et al., 2020), among other mechanisms (Korolev and Leisner, 2020).*

17. P.18.l.25: a " ," is missing between 0.03 and 5.

Corrected.

18. P.19, l.27-28: "It was found that... formation of ice particles": I find this formulation rather confusing: this work focuses on the formation of columnar ice crystals at temperatures warmer than −10 ◦C, not on the formation of ice crystals in general.

We have amended the sentence as:

*It was found that columnar ice formation is relatively frequent in clouds at temperatures of -10 ℃ or warmer.*

---

## Author Comment (AC2)

Reviewer #1

The paper is well written and contains nice results in terms of ice properties in mixed phase clouds and ice production derived from cloud radar observations. I have a small list of suggestions to improve the paper. Only minor revisions are required.

We sincerely appreciate the reviewer for the positive comments on our paper. We have amended the manuscript as suggested. Please see below our response to your comments.

Title: Two-year statistics… is more appropriate… (Feb 2018 to April 2020)
Amended
P1, l2: remote sensing IS an observation. So, remove 'observation' here, please.
We agree with the reviewer. It has been amended as 'ground-based radar observations'.
P1, l9: that that
We have amended the previous sentence, please see the revised manuscript.
P1, l15: Please use 2 K (and always Kelvin for the temperature difference throughout the paper)
Corrected
P2, l1: provide a reference please
Amended
P2, l3: I am a bit confused…, dust particles (mineral dust) are about three orders of magnitude better INPs than marine particles (sea salt) at the same temperature. Please check, McCluskey (JGR 2018, Fig 8 (marine) vs Niemand 2012 (dust) similar to DeMott, 2015, dust).
We agree with the reviewer. For a given temperature, the discrepancy caused by aerosol types can be orders of magnitude. This sentence has been amended as
*This dependence is more or less universal, but can also be affected by other factors such as the geographic location, airmass types and aerosol compositions (e.g., DeMott et al., 2010; Niemand et al., 2012; Wilson et al., 2015; DeMott et al., 2016; Petters and Wright, 2015; McCluskey et al., 2018).*
P3, l31: What is the LDR decoupling of the system? It would be interesting what is the minimum and maximum LDR that can be detected within these clouds for the usual ranges of Z, Signal and LDR.
LDR decoupling is about 30 dB, so the minimum observable LDR is about -30 dB. We have added this sentence in the revised manuscript.
In single-layer clouds, as shown in Figure 7, LDR is as high as -15 dB. However, it is more complex in multi-layer clouds, since LDR can be masked by larger particles.
The suggestion proposed by the reviewer is exactly what we are working on. The current method basically follows (Oue et al., 2015; Li and Moisseev, 2020), but we are developing a more general method to detect ice columns. It works well and we would like to introduce it to the community in an upcoming submission.
P4, l10: The website is depreciated, now it is cloudnet.fmif.fi .
Further remark: For ACTRIS/Cloudnet datasets DOIs are available. Please refer to the specific dataset that was used in this study with such a DOI.
Amended to http://cloudnet.fmi.fi/.
We have added the citation of ICON data in this section.
P4, l30: Here, the Myagkov (2016) references would be appropriate to be included:
Myagkov, A., Seifert, P., Bauer-Pfundstein, M., and Wandinger, U.: Cloud radar with hybrid mode towards estimation of shape and orientation of ice crystals, Atmos. Meas. Tech., 9, 469–489,

https://doi.org/10.5194/amt-9-469-2016, 2016.

Myagkov, A., Seifert, P., Wandinger, U., Bühl, J., and Engelmann, R.: Relationship between temperature and apparent shape of pristine ice crystals derived from polarimetric cloud radar observations during the ACCEPT campaign, Atmos. Meas. Tech., 9, 3739–3754, https://doi.org/10.5194/amt-9-3739-2016, 2016.

Agree. We have added these two papers in the literature.

P5, l3: Again a reference to Radenz et al. (2019) would be appropriate here.

Agree. We have referred to Radenz et al. (2019).

P5, Figure 1 (c), this third peak (the last one to the right) indicates the presence of liquid water. Could the columnar ice particles actually have been produced by primary ice formation in this liquid layer and not be a product of ice multiplication?

If the estimated ice number concentration is lower than or comparable with the estimated INP concentration, they may form via primary ice nucleation.

The key to differentiate ice multiplication from primary ice production is whether the ice number concentration exceeds INP concentration. We did the analysis for single-layer clouds in this study, but not for multi-layered clouds. In this figure, in the section of Methods, we want to show how Doppler spectrum can identify ice populations. Therefore, we did not identify the mechanism of columnar ice production here.

P7, Figure 2(b): These statistics are highly questionable. What do they show us? Is ICON able to derive actual realistic humidity values for the clouds under study? Did ICON even resolve the clouds under study?

As stated in the first sentence of Section Results, we present statistics of 'environmental conditions associated with columnar ice production'. Therefore, it is relevant to show temperature and relative humidity statistics, and ICON data which represent a large-scale average without small-scale variability are what we currently have. But we agree that the limitation of forecasted $RH_{liquid}$ should be explained. We have amended the description as

*However, the values of $RH_{liquid}$ and $RH_{ice}$ should be interpreted with caution. ICON applies a liquid saturation adjustment, limiting the liquid supersaturation to saturation. $RH_{liquid}$ values exceeding 100 % are attributed to numerical artifacts. $RH_{ice}$ was calculated based on the forecasted temperature, pressure as well as $RH_{liquid}$, therefore can be affected by numerical artifacts as well. Given the uncertainty of ICON forecasts, we regard the presented statistics in Fig.2 as a sanity check for our method.*

P10, Figure 6(a): The figure seems to contain mixed-phase and ice clouds. Would it be possible to show both species in this figure separately?

We did give thinking on this. It would be interesting to know whether and how the presence of supercooled liquid water is linked to ice columns. Given this work is presenting statistics, we want to keep the results as less biased as possible. The most reliable way of detecting liquid is definitely lidar, however, lidar usually only sees the lowest liquid layer. Radar data, such as Doppler spectra, have the capability to detect liquid, but the performance is still under discussion (Silber et al., 2020; Kalesse et al., 2021; Kalogeras et al., 2021; Vogl et al., 2021). Therefore, we did not do such analysis.

The excellent observation facilities at Hyytiälä do enable depth-in analysis studies on this topic, and we will have a serial of works on this. For example, our separate work published on ACP (Li et al., 2021).

Kalesse-Los, H., Schimmel, W., Luke, E. and Seifert, P., 2021. Evaluating cloud liquid detection using cloud radar Doppler spectra in

a pre-trained artificial neural network against Cloudnet liquid detection. Atmospheric Measurement Techniques Discussions, 1-19.

Kalogeras, P., Battaglia, A. and Kollias, P., 2021. Supercooled Liquid Water Detection Capabilities from Ka-Band Doppler Profiling Radars: Moment-Based Algorithm Formulation and Assessment. Remote Sensing, 13(15), 2891.

Li, H., Korolev, A. and Moisseev, D., 2021. Supercooled liquid water and secondary ice production in Kelvin–Helmholtz instability as revealed by radar Doppler spectra observations. Atmospheric Chemistry and Physics Discussions, 1-22.

Silber, I., Verlinde, J., Wen, G. and Eloranta, E.W., 2019. Can Embedded Liquid Cloud Layer Volumes Be Classified in Polar Clouds Using a Single-Frequency Zenith-Pointing Radar?. IEEE Geoscience and Remote Sensing Letters, 17(2), 222-226.

Vogl, T., Maahn, M., Kneifel, S., Schimmel, W., Moisseev, D. and Kalesse-Los, H., 2021. Using artificial neural networks to predict riming from Doppler cloud radar observations. Atmospheric Measurement Techniques Discussions, 1-26.

P10, l18: Again, using unit Kelvin (…10 K) is really important here to differentiate between temperature differences and absolute temperature values.

Corrected.

P11, Figure 7 (d): Are the LWP values reliable if precipitation (especially rain) reaches the ground? It is known that microwave radiometers have a problem with such conditions - is that also true for the radar-derived LWP?

The reviewer is correct. Before processing the data, we had excluded rainfall cases. As stated in the first paragraph of Section Results:

*Given the data selection criteria, no rainfall or summer cloud cases were analyzed.*

P16, Section 5: This chapter is rather speculative and does not show a conclusive result.

We fully agree with the reviewer. However, we want to point out that:

1) The aim of this chapter is to identify the potential mechanism of producing ice columns, not developing a method to estimate concentrations of ice or INPs.

2) The derived $N_{needle}$ is underestimated. Therefore, if $N_{INPs}$ is orders of magnitude lower than $N_{needle}$, the statement that "primary INPs are inadequate to explain $N_{needle}$" is supported. This point is conclusive and is of course important.

3) We do agree that our discussion about SIP has some speculative characteristics. Because SIP itself is a topic of interest and under discussion, and there are many unknowns about SIP to be unveiled and we could not give conclusive statements. For example, even though we know $N_{needle} \gg N_{INPs}$, we could not firmly state that whether SIP is active and which one is dominant, as discussed in the last paragraph of this chapter.

- N_neelde should be computed taking into account that the size of needles is distributed spectrally. (I think the term "characteristic needle" points to this fact.)

We agree with the reviewer. Considering the size distribution of needles gives more realistic estimate of $N_{needle}$.

In this section, we want to identify the mechanism of producing ice columns. If the SIP is active, then $N_{needle}$ should be **orders of magnitude** higher than $N_{INPs}$ (Field et al., 2017). Hence, our method does not aim to accurately retrieve $N_{needle}$, but estimate its **magnitude**. We have acknowledged this point in the text: "$m_{needle}$ would be mainly determined by larger ice particles, and therefore the resulting **$N_{needle}$ is underestimated**."

We compare the underestimated **$N_{needle}$** with parameterized $N_{INPs}$. If the former is orders of magnitude higher than the later one, then we may say that the SIP is active.

- An error analysis is necessary here. What is the impact of residual measurement errors in fall velocity on the ice crystal number concentration?

Please see our response below.

- INP is compared against ice number concentration: A thorough error analysis for both values is needed.

We agree with the reviewer that the current approach is not thorough enough. However, this study is not devoted in developing an algorithm, but to identify the mechanism of producing new ice columns. We show that $N_{needle}$, which is known to be underestimated, is 2 ~ 5 orders of magnitude higher than $N_{ice}$. Therefore, it is sufficient to support the conclusion that primary INPs are inadequate to explain the high values of $N_{needle}$, and the SIP is highly plausible.

- How is ambient INP concentration over the measurement site derived?

The ambient INP concentration was measured by Ice Nucleation Spectrometer of the Karlsruhe Institute of Technology (INSEKT) as described by Schiebel (2017). From February 2018 to June 2018 (HyICE-2018 campaign), INSEKT was deployed at Hyytiälä. The INP measurements have been parameterized, and the temperature dependence of INPs was given by Schneider et al., (2020).

Schiebel, T., 2017. Ice nucleation activity of soil dust aerosols (Doctoral dissertation, KIT-Bibliothek).

P19, l10: The last sentence of the paragraph is not clear, please explain in more detail. "For many cases" is a bit blurry.

The previous statement was not clear. We have amended it as

As shown in Fig. 12, the majority of $N_{needle}$ values fall in the range of $10^{-2} \sim 10^{1}$ $L^{-1}$, which is similar with aircraft measurements obtained in tropical stratiform clouds (Yang et al., 2020).